# PGLEARN – AN OPEN-SOURCE LEARNING TOOLKIT FOR OPTIMAL POWER FLOW

## ABSTRACT

Machine Learning (ML) techniques for Optimal Power Flow (OPF) problems have recently garnered significant attention, reflecting a broader trend of leveraging ML to approximate and/or accelerate the resolution of complex optimization problems. These developments are necessitated by the increased volatility and scale in energy production for modern and future grids. However, progress in ML for OPF is hindered by the lack of standardized datasets and evaluation metrics, from generating and solving OPF instances, to training and benchmarking machine learning models. To address this challenge, this paper introduces PGLearn, a comprehensive suite of standardized datasets and evaluation tools for ML and OPF. PGLearn provides datasets that are representative of real-life operating conditions, by explicitly capturing both global and local variability in the data generation, and by, for the first time, including time series data for several large-scale systems. In addition, it supports multiple OPF formulations, including AC, DC, and second-order cone formulations. Standardized datasets are made publicly available to democratize access to this field, reduce the burden of data generation, and enable the fair comparison of various methodologies. PGLearn also includes a robust toolkit for training, evaluating, and benchmarking machine learning models for OPF, with the goal of standardizing performance evaluation across the field. By promoting open, standardized datasets and evaluation metrics, PGLearn aims at democratizing and accelerating research and innovation in machine learning applications for optimal power flow problems.

## 1 INTRODUCTION

The rapid evolution of energy systems, driven by mass integration of renewable and distributed energy resources, is creating new challenges in the maintenance, expansion, and operation of power grids. The increased volatility and scale of power generation in modern and future grids calls for innovative solutions to manage uncertainty and ensure reliability (Zhang et al., 2021). Machine learning (ML) has emerged as a powerful tool in this context, offering the potential to address key problems such as optimizing grid operations and predicting power demand, enabling the use of previously intractable applications such as real-time risk analysis (Chen et al., 2024). However, the success of ML models depends on the availability of high-quality data, which is essential for training accurate and reliable models (Khaloie et al., 2024; Lovett et al., 2024).

Optimal Power Flow (OPF) is a fundamental problem in power systems operations, focusing on how to efficiently operate a power transmission system while satisfying physics, engineering, and operations constraints. Most market-clearing algorithms for real-time electricity markets are based on OPF, which makes it paramount to real-time operations. In addition, OPF forms the building block of security-constrained unit commitment (SCUC) formulations used in day-ahead markets (Chen et al., 2023), as well as transmission expansion planning problems. In practice, many instances of OPF need to be solved at once in order to account for uncertainties in renewable power generation and/or demand, making it a computationally intensive task – especially given the non-convex physics of AC power flow. Besides its real-world applications, OPF has also garnered significant attention as a test-bed for research methods integrating mathematical programming and machine learning.

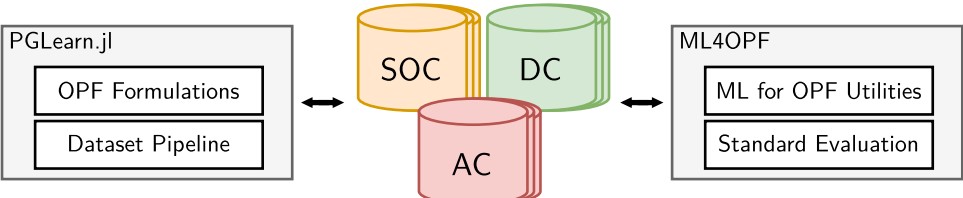

Figure 1: The PGLearn Toolkit: publicly available AC, DC, and SOC optimal power flow datasets, `PGLearn.jl` for data generation, and the `ML4OPF` ML toolkit.

The key motivation for using ML to address parametric OPF stems from the increased volatility and scale in power generation in modern and future grids. Indeed, the growing use of intermittent renewable energy sources such as wind and solar generation is driving significant growth in operational uncertainty. This motivates a shift from deterministic to, e.g., stochastic optimization formulations that explicitly consider uncertainty. Such a change results in OPF problems that are, or will be, orders of magnitude larger than today's instances. In addition, the risk of energy shortage, congestion, and voltage issues has become substantially larger and requires novel methods to manage in real-time. Machine learning offers some hope in addressing these challenges by moving much of the computational burden offline and delivering orders of magnitude speedups for real-time operations.

This research avenue is further justified by the fact that practitioners often solve OPF instances on very similar – or even identical – transmission systems, with only renewable generation and/or power demand varying across instances. Readers are referred to Hasan et al. (2020); Khaloie et al. (2024) for a detailed review of prior works in machine learning for OPF. Note that many of the works therein do not consider the non-convex AC-OPF formulation directly, but rather focus on more tractable, i.e., convex, OPF formulations, such as the DC-OPF linear approximation or second-order cone relaxation (Molzahn and Hiskens, 2019). Although these relaxed formulations do not capture the exact physics, they more closely match the problems that real power market operators and participants solve every day (Ma et al., 2009). For example, Chen (2023) uses a linear formulation inspired by problem solved by the Midcontinent Independent System Operator (MISO) to clear its real-time market.

### 1.1 Motivation: Data Scarsity in ML for OPF Research

Despite strong interest from industry, real, industry-scale data is scarce, mainly due to regulatory barriers that restrict the sharing of sensitive information on power grids. Thus, most previous ML for OPF works generate their own artificial datasets, often based on the Power Grid Lib Optimal Power Flow (PGLib-OPF) (Babaeinejadsarookolaee et al., 2019) benchmark library – a collection of grid snapshots originally designed for benchmarking AC-OPF optimization algorithms.

While machine learning methods usually require many thousands of data points to train accurate models, PGLib-OPF only provides a single snapshot per grid. Hence, a data augmentation strategy is needed to "sample around" the provided snapshots in a realistic fashion. There is however no consensus in prior literature for how to perform this sampling, which has led to a highly fragmented ecosystem where it is impossible to directly compare results from different works due to the use of very different data distributions (Khaloie et al., 2024). Indeed, the characteristics of the learning problem may vary substantially when considering different augmentation schemes, for example correlated versus uncorrelated noise.

To ensure that ML for OPF research is useful in practice, it is important to carefully consider the choices involved in designing a data augmentation scheme. For example, most works surveyed in Khaloie et al. (2024) sample instances by perturbing individual loads independently of each other. Figure 2 illustrates the limitations the resulting data distribution, for a system with 1354 buses. The figure shows that, i) the resulting distribution displays a very narrow range of total demand, and ii) the resulting OPF solutions exhibit simplistic patterns that do not capture global dynamics over a

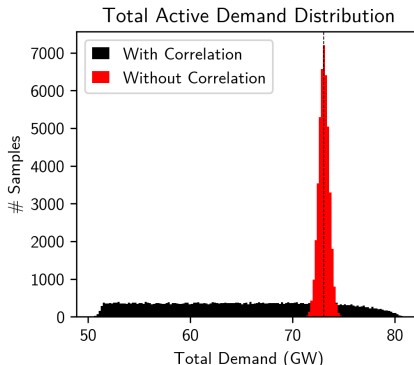 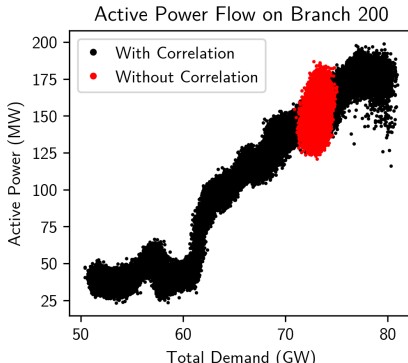

Figure 2: Limitations of sampling strategies that do not consider correlations across individual loads. Left: histogram of total demand: the absence of correlations yields a narrow range of total demand. Right: active power flow on branch 200; the absence of correlations in input data leads to datasets with low variance and diversity.

wider range of total demand. As a consequence, ML models trained on data that does not capture correlations can only be expected to perform well for a small total demand range, severely limiting their usability in practice.

Finally, the lack of publicly available standardized datasets requires individual teams to expend considerable computational resources to generate new datasets. This comes at a high financial and environmental cost, and results in most academic studies considering small, synthetic power grids which incur lower data generation costs, but are irrelevant to industry practitioners. More importantly, it represents a significant barrier to entry to teams without substantial computational resources.

## 1.2 RELATED WORK

Due to strong interest from academia and industry, several OPF datasets and data generation packages have been released in recent years. However, none simultaneously meet the requirements of being actively maintained, considering large power networks, and using realistic sampling schemes.

Some prior works (Donon et al., 2020; Chatzos et al., 2022; Klamkin et al., 2024) report results on industry-based datasets, i.e., using data obtained from transmission system operators. Although these works report more closely match real-world systems, the corresponding datasets are not publicly available due to regulations around privacy and security.

Despite a growing literature on ML for OPF, few datasets have been made publicly available. The datasets initially released alongside the OPFSampler (Robson et al., 2019) codebase are no longer available, and the code is unmaintained. The OPFLearn library (Joswig-Jones et al., 2022) provides data augmentation tools built on top of PowerModels Coffrin et al. (2018), as well as a collection of 10,000 samples of AC-OPF instances and their solution for five systems with up to 118 buses. This code is no longer maintained, only considers uncorrelated demand perturbations, and reports incomplete primal/dual solutions. More recently, and closest to this paper, OPFData (Lovett et al., 2024) is a collection of AC-OPF datasets which considers systems with up to 13,659 buses. The collection also includes instances with perturbed topology, obtained by randomly removing individual lines or generators in the system. The main limitation of OPFData, however, is that it only considers uncorrelated demand perturbation, leading to simplistic data distributions as illustrated in Figure 2. Additionally, OPFData only reports AC-OPF solutions, does not include dual solutions nor metadata such as solve times, and does not include the source code used to generate and solve samples.

## 1.3 CONTRIBUTIONS AND OUTLINE

To address the above challenges, this paper proposes PGLearn, a collection of datasets and tools for ML and OPF. The contributions of PGLearn are as follows:

- PGLearn provides a collection of standardized datasets for large-scale OPF instances, generated using a realistic and reproducible data augmentation methodology. The entire collection totals over 10,000,000 OPF samples, split between training and testing data to allow for direct comparison of results. Crucially, PGLearn is also the first OPF dataset to incorporate realistic time-series data for large-scale power systems.

- Each PGLearn dataset comprises complete primal and dual solutions for several OPF formulations, a unique feature compared to existing literature.

- PGLearn provides several evaluation metrics for objective and fair comparison of various ML methodologies for OPF problems, together with guidelines on performance benchmarking.

- The `PGLearn.jl` Julia (Bezanson et al., 2017) library containing the source code to generate the PGLearn datasets. The modular design of `PGLearn.jl` simplifies the implementation and execution of new data augmentation schemes and OPF formulations.

- The `ML4OPF` PyTorch (Ansel et al., 2024) library containing data parsers, optimized GPU-friendly implementations of the supported OPF formulations (objective, constraints, etc.), and other utilities for developing new ML methods for OPF.

The code used to generate PGLearn is fully open-source and relies only on open-source solvers, allowing to interrogate, reproduce, and extend each part of the dataset generation process. By openly distributing these tools and datasets, PGLearn seeks to lower the barrier of entry for researchers in the field, promoting innovation and accelerating the development of ML techniques for OPF and optimization more broadly.

The rest of this paper is structured as follows. Section 2 describes each OPF formulation included in PGLearn. Section 3 introduces the data augmentation procedure, and Section 4 presents relevant features of the `PGLearn.jl` and `ML4OPF` libraries. Section 5 provides recommendations for evaluation metrics and their reporting. Section 6 reviews the limitations of PGLearn, and Section 7 concludes the paper.

## 2 OPF FORMULATIONS IN PGLEARN

A unique feature of PGLearn is that it provides, for each OPF instance, solutions to several OPF formulations. This allows to compare, for the same input data, the performance of ML models trained using different formulations. Namely, PGLearn currently supports the nonlinear, non-convex AC-OPF, the second-order cone relaxation SOC-OPF, and the linear approximation DC-OPF. A brief summary of each formulation is provided below. Due to space considerations, full OPF formulations are stated in Appendix A.

### 2.1 OPF FORMULATIONS

**AC-OPF**  The AC-OPF is considered the "full" steady-state optimal power flow formulation. PGLearn uses the rectangular-power polar-voltage form, matching the `ACPPowerModel` formulation implemented in PowerModels (Coffrin et al., 2018). This formulation includes non-convex AC power flow physics to accurately model the power system. The full non-linear programming formulation is included in Model 1.

**SOC-OPF**  The SOC-OPF is a second-order-cone relaxation of the AC-OPF proposed by Jabr (2006a). The SOC-OPF better approximates the full-physics AC-OPF compared to the linear DC-OPF, but is more complicated to solve. A description of how to derive the SOC-OPF, and its full conic programming formulation, is included with Model 2.

**DC-OPF**  The DC-OPF is a sparse linear approximation to the AC-OPF (Christie et al., 2000). It is commonly used in industry to approximate AC-OPF in cases where solving AC-OPF within time constraints is intractable. Among other simplifications, it considers only active power and fixes all voltage magnitudes to 1. A list of all assumptions required to derive the DC-OPF, and its full linear programming formulation, is included with Model 4.

Table 1: Summary statistics of the PGLearn datasets.

| Case name | $|\mathcal{N}|$ | $|\mathcal{L}|$ | $|\mathcal{G}|$ | $|\mathcal{E}|$ | Total PD | Total PG | Global range |
|---|---|---|---|---|---|---|---|
| 14_ieee | 14 | 11 | 5 | 20 | 0.3 GW | 0.4 GW | 70% – 110% |
| 30_ieee | 30 | 21 | 6 | 41 | 0.3 GW | 0.4 GW | 60% – 100% |
| 57_ieee | 57 | 42 | 7 | 80 | 1.3 GW | 2 GW | 60% – 100% |
| 89_pegase | 89 | 35 | 12 | 210 | 6 GW | 10 GW | 60% – 100% |
| 118_ieee | 118 | 99 | 54 | 186 | 4 GW | 7 GW | 80% – 120% |
| 300_ieee | 300 | 201 | 69 | 411 | 24 GW | 36 GW | 60% – 100% |
| 1354_pegase | 1354 | 673 | 260 | 1991 | 73 GW | 129 GW | 70% – 110% |
| NewYork2030 | 1576 | 1446 | 323 | 2427 | 33 GW | 42 GW | 70% – 110% |
| 1888_rte | 1888 | 1000 | 290 | 2531 | 59 GW | 89 GW | 70% – 110% |
| 2869_pegase | 2869 | 1491 | 510 | 4582 | 132 GW | 231 GW | 60% – 100% |
| 6470_rte | 6470 | 3670 | 761 | 9005 | 97 GW | 118 GW | 60% – 100% |
| Texas7k | 6717 | 4541 | 637 | 9140 | 75 GW | 97 GW | 80% – 120% |
| 9241_pegase | 9241 | 4895 | 1445 | 16049 | 312 GW | 530 GW | 60% – 100% |
| 13659_pegase | 13659 | 5544 | 4092 | 20467 | 381 GW | 981 GW | 60% – 100% |
| Midwest24k | 23643 | 11727 | 5646 | 33739 | 104 GW | 318 GW | 90% – 130% |

## 2.2 Dual OPF Formulations and Solutions

Another unique feature of PGLearn is that it provides, for each OPF formulation, complete primal and dual solutions. This novel capability is motivated by the recent interest in leveraging dual information in ML contexts. For instance, Qiu et al. (2024) and Tanneau and Van Hentenryck (2024) both consider learning dual optimization proxies, wherein an ML model outputs dual-feasible solutions to conic optimization problems. In a similar fashion, Kotary and Fioretto (2024) leverage insights from Augmented Lagrangian methods to learn Lagrange multipliers for nonlinear problems. Finally, several recent works attempt to predict dual solutions in the context of mixed-integer optimization, with the aim of obtaining high-quality dual bounds through Lagrangian duality Parjadis et al. (2023); Demelas et al. (2024).

PGLearn leverages Lagrangian duality for nonlinear, non-convex problems such as AC-OPF, and conic duality for convex relaxations and approximations such as SOC-OPF and DC-OPF. Dual formulations for SOC-OPF and DC-OPF are stated in Appendix A. Note that a key advantage of conic (convex) relaxations is that dual-feasible solutions provide valid certificates of optimality, which can be used to validate the quality of primal predictions.

Dual solutions are also at the core of price formation in electricity markets. Hence, by systematically providing dual solutions, PGLearn will support future research at the intersection of optimization, machine learning, and the economics of electricity markets.

## 3 The PGLearn Collection of Datasets for Learning OPF

Like most prior work in the field, PGLearn uses a sampling scheme to convert static snapshots to datasets of OPF instances. PGLearn considers a total of 14 snapshots, split into four categories based on the number of buses:

**Small (<1k)**: 14_ieee, 30_ieee, 57_ieee, 89_pegase, 118_ieee, 300_ieee
**Medium (<5k)**: 1354_pegase, NewYork2030, 1888_rte, 2869_pegase
**Large (<10k)**: 6470_rte, Texas7k, 9241_pegase
**Extra-Large (>10k)**: 13659_pegase, Midwest24k

The 89_pegase, 1354_pegase, 2869_pegase, 9241_pegase, and 13659_pegase cases from Fliscounakis et al. (2013) are based on the European power grid, the 1888_rte and 6470_rte cases from Josz et al. (2016) are based on the French power grid, the nyiso_2030_v11 case from University of Wisconsin-Madison (2024) is based on the planned 2030 New York power grid, and the Texas7k and Midwest24k cases from Kunkolienkar et al. (2024) are based on the

Texas power grid and the US Midwest power grid, respectively. The smaller `14_ieee`, `30_ieee`, `118_ieee`, and `300_ieee` cases from University of Washington, Dept. of Electrical Engineering (1999) are synthetic. These cases are chosen to span a range of scales and to match cases used in prior ML for OPF literature.

Table 1 reports statistics of each snapshot used in the PGLearn collection. Namely, for each reference snapshot, the table reports: the number of buses $|\mathcal{N}|$, the number of loads $|\mathcal{L}|$, the number of generators $|\mathcal{G}|$, the number of branches $|\mathcal{E}|$, which includes power lines and transformers, the total active power demand in the reference case (Total PD), the total maximum active power generation (Total PG), and the range of the global scaling factor used in the data augmentation scheme described next (Global range).

**Demand Sampling** The PGLearn datasets sample each load's active and reactive demand by combining a global (per-sample) correlation term with local (per-load per-sample) noise. The power factor of each load is varied by sampling the local noise independently for the active and reactive components. This sampling procedure mimics real power system behavior since in practice, machine learning training takes time, so models are trained day-ahead on demand ranges given by forecasting systems for the next day (Chen et al., 2022). The local noise is applied to generate diverse samples at all values of total load, ensuring the usability of the machine learning system under various demand settings. The global correlation is important to ensure that the model captures a wide total demand *range* rather than being specialized to a particular total demand level. This captures more operating regimes of the power system, as shown in Figure 2. The demand sampling process is described in Algorithm 1. The Global Range column in Table 1 contains the values for $b_l$ and $b_u$ for each case. $\epsilon$ is set to 20% for all cases. The width of the global range is fixed to 40% with $b_u$ determined by incrementally scaling the reference load values in 10% steps until an infeasible case is hit.

---

**Algorithm 1** Demand Sampling

---

**Input:** Reference demand $(\mathbf{p}^{\mathrm{d}}, \mathbf{q}^{\mathrm{d}})$, global range $(b_l, b_u)$, noise level $\epsilon$
**Output:** Sampled demand $(\tilde{\mathbf{p}}^{\mathrm{d}}, \tilde{\mathbf{q}}^{\mathrm{d}})$

1: $b \sim \mathrm{Uniform}(b_l,\, b_u)$
2: **for** $i = 1 \ldots |\mathcal{L}|$ **do**
3:     $\epsilon_i^{\mathrm{p}} \sim \mathrm{Uniform}(1 - \epsilon,\, 1 + \epsilon)$
4:     $\epsilon_i^{\mathrm{q}} \sim \mathrm{Uniform}(1 - \epsilon,\, 1 + \epsilon)$
5:     $\tilde{\mathbf{p}}_i^{\mathrm{d}} \leftarrow b \cdot \epsilon_i^{\mathrm{p}} \cdot \mathbf{p}_i^{\mathrm{d}}$
6:     $\tilde{\mathbf{q}}_i^{\mathrm{d}} \leftarrow b \cdot \epsilon_i^{\mathrm{q}} \cdot \mathbf{q}_i^{\mathrm{d}}$
7: **end for**
8: **return** $(\tilde{\mathbf{p}}^{\mathrm{d}}, \tilde{\mathbf{q}}^{\mathrm{d}})$

---

**Status Sampling** To generate the $N - 1$[1] datasets, disabled branches/generators are sampled following the procedure used in OPFData (Lovett et al., 2024). Either one generator or one (non-bridge, to preserve connectedness of the network) branch is disabled per instance.

**Time-Series Sampling** There are several public resources, i.e. ENTSO-E Hirth et al. (2018) and OEDI[2], which provide time-series power demand information. Some system operators, e.g. RTE, also publish load information in real-time.[3] Although these data sources are useful for e.g. power demand forecasting studies, they are not detailed enough to formulate an OPF; namely a full description of the power system and load-level demand information is required. Motivated by this mismatch, the `Texas7k` and `Midwest24k` cases include one year of synthetic time-series data at an hourly granularity. Readers are referred to Li et al. (2020a) for details on how the coarse time-series data is created. PGLearn uses cubic spline interpolation to augment the coarse time-series in order to provide AC, DC, and SOC-OPF solutions at a five-minute granularity for `Texas7k` and ten-minute granularity for `Midwest24k`.

---

[1]$N - 1$ refers to the common security requirement that the system remain stable under any single failure; here, $N$ refers to the number of components which are susceptible to failure (branches and generators).

[2]https://data.openei.org/s3_viewer?bucket=arpa-e-perform
[3]https://www.rte-france.com/en/eco2mix/market-data

**Train-Test Split**   The training and testing sets contain only feasible input samples, i.e. those inputs for which a solution was found for all formulations. The feasible samples are then shuffled using a seeded random number generator (`MersenneTwister(42)` from Julia 1.11.5). Then, the first 80% of the shuffled feasible samples are selected as training data and the remaining 20% as testing data. Users should further split the training data to create validation and/or calibration sets as needed; the testing data should be kept unchanged to allow for direct comparison of reported results. This creates three datasets – `train`, `test`, and `infeasible`, where samples in `infeasible` are those for which a (locally) optimal solution could not be found for at least one of the formulations considered.

## 4   OPEN-SOURCE IMPLEMENTATIONS

PGLearn leverages two MIT-licensed open-source repositories: `PGLearn.jl` to generate datasets and `ML4OPF` to build machine learning models.

### 4.1   PGLEARN.JL: OPF DATA GENERATION

The `PGLearn.jl` repository contains the JuMP (Lubin et al., 2023) implementations of the OPF formulations as well as utilities for sampling and solving datasets of instances. For each case, it first uses the `make_basic_network` function from PowerModels (Coffrin et al., 2018) to parse and pre-process the corresponding raw Matpower (Zimmerman et al., 2010) file. The `MathOptSymbolicAD` (LANL-ANSI, 2022) automatic differentiation backend is used to accelerate derivative calculations. `PGLearn.jl` leverages the GNUparallel utility (Tange, 2022) for parallelization across CPU cores and the SLURM workload manager (Jette and Wickberg, 2023) for parallelization across nodes.

### 4.2   ML4OPF: MODEL TRAINING, EVALUATION AND BENCHMARKING

The `ML4OPF` library – specifically the `parsers` submodule – is the standard way to work with the PGLearn datasets. `ML4OPF` also includes several other submodules that allow researchers to quickly combine and modify existing methods, implement new methods, and easily compare results to prior works. The `layers` submodule contains implementations of several useful differentiable layers implemented in PyTorch (Ansel et al., 2024) for producing predictions which satisfy constraints. The `functional` submodule contains PyTorch JIT implementations of each formulation's constraints, objective, and incidence matrices. `formulations` contains a higher-level API which makes some common assumptions (e.g. only $\mathbf{p}^d$ and $\mathbf{q}^d$ vary per sample) to simplify common workflows. `models` contains ready-to-train implementations of various optimization proxy model architectures including the Lagrangian Dual Framework (Fioretto et al., 2021), a penalty method, and the E2ELR network from Chen (2023).

## 5   BENCHMARKING MACHINE LEARNING MODELS FOR OPF

This section describes evaluation metrics for optimization proxies, catering to the specific context of learning the solution maps of optimization problems. It also provides guidelines for comparing and benchmarking models. It is important to recognize that there is no universal metric, and that researchers should report a combination of metrics to accurately capture the behavior and performance of their models, keeping in mind the downstream use-cases of their contribution.

### 5.1   ACCURACY METRICS

The following lists several important metrics in the evaluation of optimization proxy models, i.e. models that predict the output of a parametric optimization problem given the parameters. They are stated below on a per-instance basis; aggregations should be performed by taking the mean/standard deviation and maximum (e.g. over the test set samples).

**Optimality gap**   This metric reports how close the predicted objective value (i.e. the objective value of the predicted solution) is to the true optimal value. It is important to note that predicted solutions are often infeasible, hence it is not fair to assess solutions based on optimality gap alone.

**Constraint violations**   This metric reports the magnitude of constraint violation, aggregated per group of constraints. Here, "group" refers to constraint of the same type, e.g. the $|\mathcal{N}|$ Kirchhoff's current law constraints in DC-OPF. In addition to average/maximum violation magnitude, researchers should also report (for each group of constraints) the proportion of constraints violated and the total violation (the sum of violations within each group).

**Distance to feasible set**   This metric reports how far the predicted solution is from the closest feasible point. This requires solving the corresponding projection problem for each instance (replacing the objective with $\|x - x^\star\|$ where $x^\star$ is the predicted solution and $x$ is subject to the original constraints).

**Distance to optimal solution**   This metric reports how far the predicted solution is from the optimal solution. Note that a solution can exhibit small residuals but large distance to feasibility. In real-life, this can mean that a solution with small residual may need large changes to become feasible, with potentially a large increase in cost.

## 5.2   Computational performance metrics

These metrics evaluate how fast the proxy models are, compared to optimization solvers. It's important to recognize that ML proxies are only heuristics, whereas optimization solvers have stronger guarantees. Two types of metrics should be reported: computing time for applications where a single instance is solved at a time and throughput for applications that need to solve large batches of instances (e.g. large-scale simulations). Timing results should be reported using CPU/GPU.hour (i.e. 2 CPU cores for 1 hour corresponds to 2 CPU.hr). Similarly to the accuracy metrics, aggregated timing results should report both the mean/standard deviation and the maximum across samples. Finally, in line with general guidelines for reporting ML results, researchers should always report the device (CPU and/or GPU) used for running experiments.

**Data-generation time**   This metric reports the total time spent obtaining the ground-truth primal/dual solutions required for the training (and validation) set of the proxy model – the time spent generating the test set can be excluded.

**Training time**   This metric reports the time spent training the optimization proxy model. In addition, researchers should comment on how often the model would need to be re-trained in a practical application. For instance, it is not realistic to train a model every hour if the training time is 6 hours.

**Inference time**   This metric reports the time spent producing a solution to a single instance. This is useful if the downstream application involves solving instances sequentially, for instance, when solving a market-clearing problem every 5 minutes. Researchers should report the maximum time across samples, especially for architectures that involve an iterative scheme such as gradient correction (Donti et al., 2021), implicit layers (Agrawal et al., 2019), or optimization solvers, because of performance variability.

**Instance throughput**   This metric reports how many instances can be processed per unit of time with a fixed computational resource budget. This metric is relevant for settings where a large number of instances need to be evaluated, e.g., when running large-scale simulations that require solving multiple OPF instances. In addition, it better captures the batched processing advantages of GPU devices.

The `solve_time` metadata included with the PGLearn datasets can be used to obtain data-generation time and instance throughput for optimization solvers. However, it is important to note that PGLearn datasets are generated using a single thread per instance, and utilize multiple processes per machine. Such settings are known to have an adverse impact on the performance of optimization solvers. Hence, the solving times reported in the metadata are likely over-estimates compared to solving one instance at a time in a "clean" environment or with multi-threaded solvers.

## 6 Limitations

### 6.1 Data Limitations

In the absence of publicly-available, granular datasets released directly by system operators, PGLearn is limited like prior works to using synthetic time-series and data augmentation schemes to generate samples. Nevertheless, it is important to note that the reference snapshots selected for PGLearn are based on real power grids in France, Europe, Texas, and the American midwest (Josz et al., 2016; Fliscounakis et al., 2013; Kunkolienkar et al., 2024; University of Wisconsin-Madison, 2024), and the time-series used are based on real-world characteristics Li et al. (2020b).

Another limitation of PGLearn is the limited variety of topologies, and the nature of topology changes. Namely, PGLearn considers topology variations by removing individual lines or generators. In contrast, real-life operations include multiple categories of topology changes, such as switching multiple lines and reconfiguring buses within a substation. Additional research is needed to better capture this lesser-studied facet of power grid operations.

### 6.2 Future Collections

PGLearn aims to provide curated datasets that are updated over time, to integrate new data-generation procedures and OPF formulations. To that end, future versions of PGLearn will comprise OPF formulations that include elements present in market-clearing formulations used by system operators. This includes, for instance, the integration of reserve products and support for piece-wise linear production curves Ma et al. (2009).

## 7 Conclusion

This paper has introduced PGLearn, an open-source learning toolkit for optimal power flow. PGLearn addresses the lack of standardized datasets for ML and OPF by releasing several datasets of large-scale OPF instances. It is the first collection that comprises complete primal and dual solutions for multiple OPF formulations. In addition to releasing public datasets, PGLearn provides open-source tools for data generation, and for the training and evaluation of ML models. These open-source tools enable reproducible and fair evaluation of methods for ML and OPF, thereby democratizing access to the field.

The PGLearn collection contains, in its initial release, over 10,000,000 OPF samples. It is released alongside extensive documentation and code, allowing users to generate additional datasets as needed, and to benchmark the performance of ML models. The paper has also provided several performance metrics and guidelines on how to report them. These guidelines aim at capturing specific aspects of ML for OPF that fall outside the scope of traditional ML applications. This includes, for instance, the fundamental importance of measuring and reporting constraint violations, as well as accurate reporting of data-generation, training and inference times when evaluating computational performance.

Finally, PGLearn aims to democratize access to research on ML and OPF by removing the barrier to entry caused by the computational requirements of large-scale data generation. It also aims to align academic research more closely to the scale and complexity of real-world power systems. This will, in turn, unlock the potential for modern AI techniques to assist in making future energy systems more efficient, reliable, and sustainable.

### Reproducibility Statement

Since the entire pipeline for generating the PGLearn datasets is open-source, the dataset is completely reproducible. The Julia random number generator `MersenneTwister` is used to ensure random number generation is consistent across machines. The `ML4OPF` repository similarly makes use of seeded random number generators, e.g. when instantiating neural network weights and shuffling training data. Besides allowing to fully recreate the PGLearn datasets, the focus on reproducibility also allows practitioners to easily extend or modify the dataset, for example generating new datasets based on custom formulations.

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

# A FORMULATIONS

## A.1 BACKGROUND MATERIAL

This section provides a brief overview of Lagrangian and conic duality. PGLearn uses the former for nonlinear non-convex problems such as AC-OPF, and the latter for convex formulations such as SOC-OPF and DC-OPF.

### A.1.1 NONLINEAR OPTIMIZATION

Consider a nonlinear, non-convex optimization problem of the form

$$\min_{x} \quad f(x) \tag{1a}$$

$$\text{s.t.} \quad g(x) \geq 0 \tag{1b}$$

$$h(x) = 0 \tag{1c}$$

where $f : \mathbb{R}^n \mapsto \mathbb{R}$, $g : \mathbb{R}^n \mapsto \mathbb{R}^m$ and $h : \mathbb{R}^n \mapsto \mathbb{R}^p$ are continuous functions, assumed to be differentiable over their respective domains.

Denote by $\mu \in \mathbb{R}^m$ and $\lambda \in \mathbb{R}^p$ the Lagrange multipliers associated to constraints (1b) and (1c), respectively. The first order Karush-Kuhn-Tucker optimality conditions read

$$J_h(x)^\top \lambda + J_g(x)^\top \mu = \nabla_x f(x) \tag{2a}$$

$$g(x) \geq 0 \tag{2b}$$

$$h(x) = 0 \tag{2c}$$

$$\mu \geq 0 \tag{2d}$$

$$\mu^\top g(x) = 0 \tag{2e}$$

where $J_h(x) = \nabla_x h(x)$ and $J_g(x) = \nabla_x g(x)$ denote the Jacobian matrices of $h$ and $g$, respectively.

Given Lagrange multipliers $\lambda, \mu$, the following Lagrangian bound is a valid lower bound on the optimal value of problem (1):

$$\mathcal{L}(\lambda, \mu) = \min_{x} f(x) - \lambda^\top h(x) - \mu^\top g(x). \tag{3}$$

Note that computing this Lagrangian bound requires solving a nonlinear, non-convex problem, which is NP-hard in general. Hence, it is generally intractable to compute valid dual bounds from Lagrangian duality in the context of non-convex problems.

### A.1.2 CONIC OPTIMIZATION

Consider a conic optimization problem of the form

$$\min_{x} \quad c^\top x \tag{4a}$$

$$\text{s.t.} \quad Ax \succeq_{\mathcal{K}} b \tag{4b}$$

where $A \in \mathbb{R}^{m \times n}$ and $\mathcal{K}$ is a proper cone, i.e., a closed, pointed, convex cone with non-empty interior. The corresponding conic dual problem reads

$$\max_{y} \quad b^\top y \tag{5a}$$

$$\text{s.t.} \quad A^\top y = c \tag{5b}$$

$$y \in \mathcal{K}^* \tag{5c}$$

where $\mathcal{K}^*$ is the dual cone of $\mathcal{K}$. The reader is referred to Ben-Tal and Nemirovski (2001) for a more complete overview of conic optimization and duality.

As shown by Tanneau and Van Hentenryck (2024), in many real-life applications, it is straightforward to obtain dual-feasible solutions. This is the case, for instance, when all primal variables have finite lower and upper bounds, as is the case for all formulations considered in this work. By weak conic duality, such dual-feasible solutions then yield valid dual bounds.

## A.2 OPF FORMULATIONS

This section presents the optimization models for each OPF formulation in PGLearn. Readers are referred to the Matpower manual (Zimmerman and Murillo-Sánchez, 2024) for a general introduction to power systems, as well as the underlying concepts and relevant notations. Readers are also referred to the `build_opf` functions in the `PGLearn.jl` source code for implementations of each using the JuMP (Lubin et al., 2023) modeling language, and to the `extract_primal` and `extract_dual` functions for how primal and dual solutions are extracted and stored.

To formulate OPF problems, the following sets are introduced. The set of buses is denoted by $\mathcal{N}$. The sets of generators and loads attached to bus $i \in \mathcal{N}$ are denoted by $\mathcal{G}_i$ and $\mathcal{L}_i$, respectively. The set of branches, i.e., power lines and transformers, is denoted by $\mathcal{E}$. Each edge $e \in \mathcal{E}$ is associated with a pair of buses $(i, j)$ corresponding to the edge's origin and destination. Note that power grids often include parallel branches, i.e., two branches may have identical endpoints. For ease of reading, using a slight abuse of notation, edges are identified with their endpoints using the notation $e = (i, j) \in \mathcal{E}$; this indicates that branch $e$ has endpoints $i, j$. The set of edges leaving (resp. entering) bus $i \in \mathcal{N}$ is denoted by $\mathcal{E}_i$ (resp. $\mathcal{E}_i^R$). Finally, each branch $e$ is characterized by its complex admittance matrix

$$Y_e = \begin{pmatrix} Y_e^{\text{ff}} & Y_e^{\text{ft}} \\ Y_e^{\text{tf}} & Y_e^{\text{tt}} \end{pmatrix} = \begin{pmatrix} g_e^{\text{ff}} + \mathbf{j}b_e^{\text{ff}} & g_e^{\text{ft}} + \mathbf{j}b_e^{\text{ft}} \\ g_e^{\text{tf}} + \mathbf{j}b_e^{\text{tf}} & g_e^{\text{tt}} + \mathbf{j}b_e^{\text{tt}} \end{pmatrix} \in \mathbb{C}^{2 \times 2} \tag{6}$$

where $\mathbf{j}$ is the imaginary unit, i.e., $\mathbf{j}^2 = -1$.

### A.2.1 AC OPTIMAL POWER FLOW

Model 1 states the nonlinear programming formulation of AC-OPF used in PGLearn.

---

**Model 1** AC Optimal Power Flow (AC-OPF)

$$\min_{\mathbf{p}^{\text{g}}, \mathbf{q}^{\text{g}}, \mathbf{p}^{\text{f}}, \mathbf{q}^{\text{f}}, \mathbf{p}^{\text{t}}, \mathbf{q}^{\text{t}}, \mathbf{v}, \boldsymbol{\theta}} \quad \sum_{i \in \mathcal{N}} \sum_{j \in \mathcal{G}_i} c_j \mathbf{p}_j^{\text{g}} \tag{7a}$$

$$\text{s.t.} \quad \sum_{j \in \mathcal{G}_i} \mathbf{p}_j^{\text{g}} - \sum_{j \in \mathcal{L}_i} \mathbf{p}_j^{\text{d}} - g_i^{\text{s}} \mathbf{v}_i^2 = \sum_{e \in \mathcal{E}_i} \mathbf{p}_e^{\text{f}} + \sum_{e \in \mathcal{E}_i^R} \mathbf{p}_e^{\text{t}} \qquad \forall i \in \mathcal{N} \tag{7b}$$

$$\sum_{j \in \mathcal{G}_i} \mathbf{q}_j^{\text{g}} - \sum_{j \in \mathcal{L}_i} \mathbf{q}_j^{\text{d}} + b_i^{\text{s}} \mathbf{v}_i^2 = \sum_{e \in \mathcal{E}_i} \mathbf{q}_e^{\text{f}} + \sum_{e \in \mathcal{E}_i^R} \mathbf{q}_e^{\text{t}} \qquad \forall i \in \mathcal{N} \tag{7c}$$

$$\mathbf{p}_e^{\text{f}} = g_e^{\text{ff}} \mathbf{v}_i^2 + g_e^{\text{ft}} \mathbf{v}_i \mathbf{v}_j \cos(\boldsymbol{\theta}_i - \boldsymbol{\theta}_j) + b_e^{\text{ft}} \mathbf{v}_i \mathbf{v}_j \sin(\boldsymbol{\theta}_i - \boldsymbol{\theta}_j) \quad \forall e = (i, j) \in \mathcal{E} \tag{7d}$$

$$\mathbf{q}_e^{\text{f}} = -b_e^{\text{ff}} \mathbf{v}_i^2 - b_e^{\text{ft}} \mathbf{v}_i \mathbf{v}_j \cos(\boldsymbol{\theta}_i - \boldsymbol{\theta}_j) + g_e^{\text{ft}} \mathbf{v}_i \mathbf{v}_j \sin(\boldsymbol{\theta}_i - \boldsymbol{\theta}_j) \quad \forall e = (i, j) \in \mathcal{E} \tag{7e}$$

$$\mathbf{p}_e^{\text{t}} = g_e^{\text{tt}} \mathbf{v}_j^2 + g_e^{\text{tf}} \mathbf{v}_i \mathbf{v}_j \cos(\boldsymbol{\theta}_i - \boldsymbol{\theta}_j) - b_e^{\text{tf}} \mathbf{v}_i \mathbf{v}_j \sin(\boldsymbol{\theta}_i - \boldsymbol{\theta}_j) \quad \forall e = (i, j) \in \mathcal{E} \tag{7f}$$

$$\mathbf{q}_e^{\text{t}} = -b_e^{\text{tt}} \mathbf{v}_j^2 - b_e^{\text{tf}} \mathbf{v}_i \mathbf{v}_j \cos(\boldsymbol{\theta}_i - \boldsymbol{\theta}_j) - g_e^{\text{tf}} \mathbf{v}_i \mathbf{v}_j \sin(\boldsymbol{\theta}_i - \boldsymbol{\theta}_j) \quad \forall e = (i, j) \in \mathcal{E} \tag{7g}$$

$$(\mathbf{p}_e^{\text{f}})^2 + (\mathbf{q}_e^{\text{f}})^2 \leq \overline{S_e}^2 \qquad \forall e \in \mathcal{E} \tag{7h}$$

$$(\mathbf{p}_e^{\text{t}})^2 + (\mathbf{q}_e^{\text{t}})^2 \leq \overline{S_e}^2 \qquad \forall e \in \mathcal{E} \tag{7i}$$

$$\underline{\Delta}\theta_e \leq \boldsymbol{\theta}_i - \boldsymbol{\theta}_j \leq \overline{\Delta}\theta_e \qquad \forall e = (i, j) \in \mathcal{E} \tag{7j}$$

$$\boldsymbol{\theta}_{\text{ref}} = 0 \tag{7k}$$

$$\underline{\mathbf{p}}_i^{\text{g}} \leq \mathbf{p}_i^{\text{g}} \leq \overline{\mathbf{p}}_i^{\text{g}} \qquad \forall i \in \mathcal{G} \tag{7l}$$

$$\underline{\mathbf{q}}_i^{\text{g}} \leq \mathbf{q}_i^{\text{g}} \leq \overline{\mathbf{q}}_i^{\text{g}} \qquad \forall i \in \mathcal{G} \tag{7m}$$

$$\underline{\mathbf{v}_i} \leq \mathbf{v}_i \leq \overline{\mathbf{v}_i} \qquad \forall i \in \mathcal{N} \tag{7n}$$

$$-\overline{S_e} \leq \mathbf{p}_e^{\text{f}} \leq \overline{S_e} \qquad \forall e \in \mathcal{E} \tag{7o}$$

$$-\overline{S_e} \leq \mathbf{q}_e^{\text{f}} \leq \overline{S_e} \qquad \forall e \in \mathcal{E} \tag{7p}$$

$$-\overline{S_e} \leq \mathbf{p}_e^{\text{t}} \leq \overline{S_e} \qquad \forall e \in \mathcal{E} \tag{7q}$$

$$-\overline{S_e} \leq \mathbf{q}_e^{\text{t}} \leq \overline{S_e} \qquad \forall e \in \mathcal{E} \tag{7r}$$

---

The decision variables comprise active and reactive power dispatch $\mathbf{p}^g$ and $\mathbf{q}^g$, active and reactive power flows in the forward and reverse direction $\mathbf{p}^f, \mathbf{q}^f, \mathbf{p}^t, \mathbf{q}^t$, and voltage magnitude and angle $\mathbf{v}$ and $\boldsymbol{\theta}$, respectively. The objective (7a) minimizes the production costs; PGLearn currently supports linear objective functions. Constraints (7b) and (7c) encode the active and reactive power balance physics constraints given by Kirchhoff's current law. Constraints (7d)-(7g) express active and reactive power flow following Ohm's law. Constraints (7h)-(7i) and (7j) enforce the engineering limits of the transmission lines, namely, their thermal capacity and maximum voltage angle difference. Constraint (7k) fixes the reference bus' (slack bus) voltage angle to zero. Finally, constraints (7l) and (7m) encode each generator's minimum and maximum active and reactive power outputs, constraint (7n) enforces the voltage magnitude bounds, and constraints (7o), (7p), (7q), (7r) enforce bounds on the power flow variables.

PGLearn supports any nonlinear optimization solver supported by JuMP. The default configuration solves AC-OPF instances using Ipopt (Biegler and Zavala, 2009) with the MA27 linear solver (Duff and Reid, 1982) via LibHSL (Fowkes et al., 2024). Note that, unless a global optimization solver is used, global optimality of AC-OPF solutions is not guaranteed. Nevertheless, previous experience suggests that solutions obtained by Ipopt are typically close to optimal Gopinath et al. (2020).

### A.2.2 SOC OPTIMAL POWER FLOW

PGLearn also implements the second-order cone relaxation of AC-OPF proposed by Jabr in (Jabr, 2006b; 2007), herein referred to as SOC-OPF. This convex relaxation can be solved in polynomial time using, e.g., an interior-point algorithm, and is exact on radial networks Molzahn and Hiskens (2019).

The SOC-OPF relaxation is obtained by introducing variables

$$\mathbf{w}_i = \mathbf{v}_i^2 \tag{8a}$$

$$\mathbf{w}_e^{re} = \mathbf{v}_i\mathbf{v}_j \cos(\boldsymbol{\theta}_i - \boldsymbol{\theta}_j) \tag{8b}$$

$$\mathbf{w}_e^{im} = \mathbf{v}_i\mathbf{v}_j \sin(\boldsymbol{\theta}_i - \boldsymbol{\theta}_j) \tag{8c}$$

together with the valid (non-convex) constraint

$$(\mathbf{w}_e^{re})^2 + (\mathbf{w}_e^{im})^2 = \mathbf{w}_i\mathbf{w}_j, \quad \forall e = (i,j) \in \mathcal{E}. \tag{9}$$

Then, constraints (7b), (7c), (7d), (7e), (7f), (7g), (7j), and (7n) are reformulated using these new variables, and constraint (9) is convexified into the so-called Jabr inequality

$$(\mathbf{w}_e^{re})^2 + (\mathbf{w}_e^{im})^2 \leq \mathbf{w}_i\mathbf{w}_j, \quad \forall e = (i,j) \in \mathcal{E}. \tag{10}$$

Finally, valid lower and upper bounds for $\mathbf{w}^{re}, \mathbf{w}^{im}$ variables are derived from (7n), (7j) and (9), using the same strategy as Coffrin et al. (2018). Model 2 states the resulting SOC-OPF formulation, in conic form.

The implementation in PGLearn slighly differs from Coffrin et al. (2018) in the definition of $\mathbf{w}^{re}, \mathbf{w}^{im}$ variables. Namely, in Coffrin et al. (2018), $\mathbf{w}^{re}, \mathbf{w}^{im}$ variables are defined per *bus-pair*, defined as a pair of buses $(i,j)$ linked by at least one branch $e = (i,j) \in \mathcal{E}$. In contrast, PGLearn defines $\mathbf{w}^{re}, \mathbf{w}^{im}$ variables for each branch; the two formulations are equivalent unless parallel branches are present. This design choice was motivated by the simplicity of the branch-level formulation and the corresponding data formats, and was found to have a marginal impact on the quality of the relaxation.

The dual SOC-OPF model is stated in Model 3. Dual variables $\lambda^p, \lambda^q, \lambda^{pf}, \lambda^{qf}, \lambda^{pt}, \lambda^{qt}$ are associated to equality constraints (11b), (11c), (11d), (11e), (11f), (11e), and are therefore unrestricted. Dual conic variables $\nu^f, \nu^t$ and $\omega$ are associated to conic constraints (11h), (11i) and (11j), respectively. Dual variables $\underline{\mu}^\theta$ and $\bar{\mu}^\theta$ are associated to the lower and upper side of voltage angle difference constraint (11k). Finally, dual variables $\underline{\mu}^w, \bar{\mu}^w, \underline{\mu}^{pg}, \bar{\mu}^{pg}, \underline{\mu}^{qg}, \bar{\mu}^{qg}, \underline{\mu}^{pf}, \bar{\mu}^{pf}, \underline{\mu}^{qf}, \bar{\mu}^{qf}, \underline{\mu}^{pt}, \bar{\mu}^{pt}, \underline{\mu}^{qt}, \bar{\mu}^{qt}, \underline{\mu}^{wr}, \bar{\mu}^{wr}, \underline{\mu}^{wi}, \bar{\mu}^{wi}$ are associated to lower and upper bounds on variables $\mathbf{w}, \mathbf{p}^g, \mathbf{q}^g, \mathbf{p}^f, \mathbf{q}^f, \mathbf{p}^t, \mathbf{q}^t, \mathbf{w}^{re}, \mathbf{w}^{im}$, respectively.

Note that users need not interact with the dual SOC-OPF problem directly, as interior-point solvers typically report both primal and dual information. The formulation in Model 3 is stated for complete-

**Model 2** SOC Optimal Power Flow (SOC-OPF)

$$\min_{\mathbf{p}^g,\mathbf{q}^g,\mathbf{p}^f,\mathbf{q}^f,\mathbf{p}^t,\mathbf{q}^t,\mathbf{w},\mathbf{w}^{re},\mathbf{w}^{im}} \sum_{i\in\mathcal{N}}\sum_{j\in\mathcal{G}_i} c_j \mathbf{p}_j^g \tag{11a}$$

$$\text{s.t.} \quad \sum_{j\in\mathcal{G}_i}\mathbf{p}_j^g - \sum_{j\in\mathcal{L}_i}\mathbf{p}_j^d - g_i^s\mathbf{w}_i = \sum_{e\in\mathcal{E}_i}\mathbf{p}_e^f + \sum_{e\in\mathcal{E}_i^R}\mathbf{p}_e^t \qquad \forall i\in\mathcal{N} \tag{11b}$$

$$\sum_{j\in\mathcal{G}_i}\mathbf{q}_j^g - \sum_{j\in\mathcal{L}_i}\mathbf{q}_j^d + b_i^s\mathbf{w}_i = \sum_{e\in\mathcal{E}_i}\mathbf{q}_e^f + \sum_{e\in\mathcal{E}_i^R}\mathbf{q}_e^t \qquad \forall i\in\mathcal{N} \tag{11c}$$

$$\mathbf{p}_e^f = g_e^{ff}\mathbf{w}_i + g_e^{ft}\mathbf{w}_e^{re} + b_e^{ft}\mathbf{w}_e^{im} \qquad \forall e=(i,j)\in\mathcal{E} \tag{11d}$$

$$\mathbf{q}_e^f = -b_e^{ff}\mathbf{w}_i - b_e^{ft}\mathbf{w}_e^{re} + g_e^{ft}\mathbf{w}_e^{im} \qquad \forall e=(i,j)\in\mathcal{E} \tag{11e}$$

$$\mathbf{p}_e^t = g_e^{tt}\mathbf{w}_j + g_e^{tf}\mathbf{w}_e^{re} - b_e^{tf}\mathbf{w}_e^{im} \qquad \forall e=(i,j)\in\mathcal{E} \tag{11f}$$

$$\mathbf{q}_e^t = -b_e^{tt}\mathbf{w}_j - b_e^{tf}\mathbf{w}_e^{re} - g_e^{tf}\mathbf{w}_e^{im} \qquad \forall e=(i,j)\in\mathcal{E} \tag{11g}$$

$$(\overline{S_e},\ \mathbf{p}_e^f,\ \mathbf{q}_e^f)\in\mathcal{Q}^3 \qquad \forall e\in\mathcal{E} \tag{11h}$$

$$(\overline{S_e},\ \mathbf{p}_e^t,\ \mathbf{q}_e^t)\in\mathcal{Q}^3 \qquad \forall e\in\mathcal{E} \tag{11i}$$

$$\left(\frac{\mathbf{w}_i}{\sqrt{2}},\ \frac{\mathbf{w}_j}{\sqrt{2}},\ \mathbf{w}_e^{re},\ \mathbf{w}_e^{im}\right)\in\mathcal{Q}_r^4 \qquad \forall e=(i,j)\in\mathcal{E} \tag{11j}$$

$$\tan(\underline{\Delta}\theta_e)\mathbf{w}_e^{re} \le \mathbf{w}_e^{im} \le \tan(\overline{\Delta}\theta_e)\mathbf{w}_e^{re} \qquad \forall e\in\mathcal{E} \tag{11k}$$

$$(\underline{\mathbf{v}}_i)^2 \le \mathbf{w}_i \le (\overline{\mathbf{v}}_i)^2 \qquad \forall i\in\mathcal{N} \tag{11l}$$

$$\underline{\mathbf{p}_i^g} \le \mathbf{p}_i^g \le \overline{\mathbf{p}_i^g} \qquad \forall i\in\mathcal{G} \tag{11m}$$

$$\underline{\mathbf{q}_i^g} \le \mathbf{q}_i^g \le \overline{\mathbf{q}_i^g} \qquad \forall i\in\mathcal{G} \tag{11n}$$

$$-\overline{S_e} \le \mathbf{p}_e^f \le \overline{S_e} \qquad \forall e\in\mathcal{E} \tag{11o}$$

$$-\overline{S_e} \le \mathbf{q}_e^f \le \overline{S_e} \qquad \forall e\in\mathcal{E} \tag{11p}$$

$$-\overline{S_e} \le \mathbf{p}_e^t \le \overline{S_e} \qquad \forall e\in\mathcal{E} \tag{11q}$$

$$-\overline{S_e} \le \mathbf{q}_e^t \le \overline{S_e} \qquad \forall e\in\mathcal{E} \tag{11r}$$

$$\underline{\mathbf{w}}_e^{re} \le \mathbf{w}_e^{re} \le \overline{\mathbf{w}}_e^{re} \qquad \forall e\in\mathcal{E} \tag{11s}$$

$$\underline{\mathbf{w}}_e^{im} \le \mathbf{w}_e^{im} \le \overline{\mathbf{w}}_e^{im} \qquad \forall e\in\mathcal{E} \tag{11t}$$

ness and to support research on predicting dual solutions. PGLearn supports the use of any JuMP-supported conic solver. By default, PGLearn solves SOC-OPF instances using Clarabel (Goulart and Chen, 2024).

**Model 3** Dual of SOC-OPF

$$\max_{\lambda,\mu,\nu,\omega} \quad \sum_{i\in\mathcal{N}} \left( \lambda_i^{\mathrm{p}} \sum_{j\in\mathcal{L}_i} \left(\mathbf{p}_j^{\mathrm{d}}\right) + \lambda_i^{\mathrm{q}} \sum_{j\in\mathcal{L}_i} \left(\mathbf{q}_j^{\mathrm{d}}\right) + \sum_{j\in\mathcal{G}_i} \left( \underline{\mathbf{p}}_j^{\mathrm{g}} \underline{\mu}_j^{\mathrm{pg}} + \overline{\mathbf{p}}_j^{\mathrm{g}} \bar{\mu}_j^{\mathrm{pg}} + \underline{\mathbf{q}}_j^{\mathrm{g}} \underline{\mu}_j^{\mathrm{qg}} + \overline{\mathbf{q}}_j^{\mathrm{g}} \bar{\mu}_j^{\mathrm{qg}} \right) + \underline{\mathrm{v}}_i^2 \underline{\mu}_i^{\mathrm{w}} + \bar{\mathrm{v}}_i^2 \bar{\mu}_i^{\mathrm{w}} \right)$$

$$+ \sum_{e\in\mathcal{E}} \left( -\bar{s}_e \left( \underline{\mu}_e^{\mathrm{pf}} - \bar{\mu}_e^{\mathrm{pf}} + \underline{\mu}_e^{\mathrm{qf}} - \bar{\mu}_e^{\mathrm{qf}} + \underline{\mu}_e^{\mathrm{pt}} - \bar{\mu}_e^{\mathrm{pt}} + \underline{\mu}_e^{\mathrm{qt}} - \bar{\mu}_e^{\mathrm{qt}} + \nu_e^{\mathrm{sf}} + \nu_e^{\mathrm{st}} \right) + \underline{\mathbf{w}}_e^{\mathrm{re}} \underline{\mu}_e^{\mathrm{wr}} + \overline{\mathbf{w}}_e^{\mathrm{re}} \bar{\mu}_e^{\mathrm{wr}} + \underline{\mathbf{w}}_e^{\mathrm{im}} \underline{\mu}_e^{\mathrm{wi}} + \overline{\mathbf{w}}_e^{\mathrm{im}} \bar{\mu}_e^{\mathrm{wi}} \right) \quad (12\mathrm{a})$$

$$\text{s.t.} \quad \lambda_i^{\mathrm{p}} + \underline{\mu}_g^{\mathrm{pg}} + \bar{\mu}_g^{\mathrm{pg}} = c_g \qquad\qquad \forall i\in\mathcal{N}, \forall g\in\mathcal{G}_i \quad (12\mathrm{b})$$

$$\lambda_i^{\mathrm{q}} + \underline{\mu}_g^{\mathrm{qg}} + \bar{\mu}_g^{\mathrm{qg}} = 0 \qquad\qquad \forall i\in\mathcal{N}, \forall g\in\mathcal{G}_i \quad (12\mathrm{c})$$

$$-\lambda_i^{\mathrm{p}} - \lambda_e^{\mathrm{pf}} + \nu_e^{\mathrm{pf}} + \underline{\mu}_e^{\mathrm{pf}} + \bar{\mu}_e^{\mathrm{pf}} = 0 \qquad\qquad \forall e=(i,j)\in\mathcal{E} \quad (12\mathrm{d})$$

$$-\lambda_i^{\mathrm{q}} - \lambda_e^{\mathrm{qf}} + \nu_e^{\mathrm{qf}} + \underline{\mu}_e^{\mathrm{qf}} + \bar{\mu}_e^{\mathrm{qf}} = 0 \qquad\qquad \forall e=(i,j)\in\mathcal{E} \quad (12\mathrm{e})$$

$$-\lambda_j^{\mathrm{p}} - \lambda_e^{\mathrm{pt}} + \nu_e^{\mathrm{pt}} + \underline{\mu}_e^{\mathrm{pt}} + \bar{\mu}_e^{\mathrm{pt}} = 0 \qquad\qquad \forall e=(i,j)\in\mathcal{E} \quad (12\mathrm{f})$$

$$-\lambda_j^{\mathrm{q}} - \lambda_e^{\mathrm{qt}} + \nu_e^{\mathrm{qt}} + \underline{\mu}_e^{\mathrm{qt}} + \bar{\mu}_e^{\mathrm{qt}} = 0 \qquad\qquad \forall e=(i,j)\in\mathcal{E} \quad (12\mathrm{g})$$

$$-g_i^s \lambda_i^{\mathrm{p}} + b_i^s \lambda_i^{\mathrm{q}} + \sum_{e\in\mathcal{E}_i^+} \left( g_e^{\mathrm{ff}} \lambda_e^{\mathrm{pf}} - b_e^{\mathrm{ff}} \lambda_e^{\mathrm{qf}} + \frac{\omega_e^{\mathrm{f}}}{\sqrt{2}} \right) + \sum_{e\in\mathcal{E}_i^-} \left( g_e^{\mathrm{tt}} \lambda_e^{\mathrm{pt}} - b_e^{\mathrm{tt}} \lambda_e^{\mathrm{qt}} + \frac{\omega_e^{\mathrm{t}}}{\sqrt{2}} \right) + \underline{\mu}_i^{\mathrm{w}} + \bar{\mu}_i^{\mathrm{w}} = 0 \qquad \forall i\in\mathcal{N} \quad (12\mathrm{h})$$

$$g_e^{\mathrm{ft}} \lambda_e^{\mathrm{pf}} + g_e^{\mathrm{tf}} \lambda_e^{\mathrm{pt}} - b_e^{\mathrm{ft}} \lambda_e^{\mathrm{qf}} - b_e^{\mathrm{tf}} \lambda_e^{\mathrm{qt}} - \tan(\underline{\Delta}\theta_e) \underline{\mu}_e^{\theta} + \tan(\overline{\Delta}\theta_e) \bar{\mu}_e^{\theta} + \omega_e^{\mathrm{re}} + \underline{\mu}_e^{\mathrm{wr}} + \bar{\mu}_e^{\mathrm{wr}} = 0 \qquad \forall e\in\mathcal{E} \quad (12\mathrm{i})$$

$$b_e^{\mathrm{ft}} \lambda_e^{\mathrm{pf}} - b_e^{\mathrm{tf}} \lambda_e^{\mathrm{pt}} + g_e^{\mathrm{ft}} \lambda_e^{\mathrm{qf}} - g_e^{\mathrm{tf}} \lambda_e^{\mathrm{qt}} + \underline{\mu}_e^{\theta} - \bar{\mu}_e^{\theta} + \omega_e^{\mathrm{im}} + \underline{\mu}_e^{\mathrm{wi}} + \bar{\mu}_e^{\mathrm{wi}} = 0 \qquad \forall e\in\mathcal{E} \quad (12\mathrm{j})$$

$$\nu_e^{\mathrm{f}} = (\nu_e^{\mathrm{sf}}, \nu_e^{\mathrm{pf}}, \nu_e^{\mathrm{qf}}) \in \mathcal{Q}^3, \nu_e^{\mathrm{t}} = (\nu_e^{\mathrm{st}}, \nu_e^{\mathrm{pt}}, \nu_e^{\mathrm{qt}}) \in \mathcal{Q}^3 \qquad\qquad \forall e\in\mathcal{E} \quad (12\mathrm{k})$$

$$\omega_e = \left( \omega_e^{\mathrm{f}}, \omega_e^{\mathrm{t}}, \omega_e^{\mathrm{re}}, \omega_e^{\mathrm{im}} \right) \in \mathcal{Q}_r^4 \qquad\qquad \forall e\in\mathcal{E} \quad (12\mathrm{l})$$

$$\underline{\mu}^{\mathrm{pg}}, \underline{\mu}^{\mathrm{qg}}, \underline{\mu}^{\mathrm{w}}, \underline{\mu}^{\theta}, \underline{\mu}^{\mathrm{pf}}, \underline{\mu}^{\mathrm{qf}}, \underline{\mu}^{\mathrm{pt}}, \underline{\mu}^{\mathrm{qt}} \geq 0 \qquad\qquad (12\mathrm{m})$$

$$\bar{\mu}^{\mathrm{pg}}, \bar{\mu}^{\mathrm{qg}}, \bar{\mu}^{\mathrm{w}}, \bar{\mu}^{\theta}, \bar{\mu}^{\mathrm{pf}}, \bar{\mu}^{\mathrm{qf}}, \bar{\mu}^{\mathrm{pt}}, \bar{\mu}^{\mathrm{qt}} \leq 0 \qquad\qquad (12\mathrm{n})$$

### A.2.3 DC OPTIMAL POWER FLOW

The DC-OPF is a popular linear approximation of AC-OPF, which underlies most electricity markets. The DC approximation is motivated by several assumptions, whose validity mainly holds for transmission systems. Namely, the DC approximation assumes that all voltage magnitudes are fixed to one per-unit, voltage angles are assumed to be small (i.e. $\sin(\theta) \approx \theta$), and that reactive power and power losses can be neglected. The reader is referred to Molzahn and Hiskens (2019) for additional background on the DC approximation.

Model 4 states the DC-OPF formulation used in PGLearn. Constraint (13b) enforces nodal power balance through Kirchhoff's current law. Constraint (13c) expresses active power flows on each branch according to Ohm's law, and constraint (13d) restricts the voltage angle difference between each branch's endpoints. Constraint (13e) fixes the reference bus' voltage angle to zero. Finally, constraints (13f) and (13g) enforce lower and upper limits on active power generation and active power flows.

---

**Model 4** DC Optimal Power Flow (DC-OPF)

$$\min_{\mathbf{p}^{\mathrm{g}}, \mathbf{p}^{\mathrm{f}}, \boldsymbol{\theta}} \quad \sum_{i \in \mathcal{N}} \sum_{j \in \mathcal{G}_i} c_j \mathbf{p}_j^{\mathrm{g}} \tag{13a}$$

$$\text{s.t.} \quad \sum_{j \in \mathcal{G}_i} \mathbf{p}_j^{\mathrm{g}} - \sum_{e \in \mathcal{E}_i} \mathbf{p}_e^{\mathrm{f}} + \sum_{e \in \mathcal{E}_i^R} \mathbf{p}_e^{\mathrm{f}} = \sum_{j \in \mathcal{L}_i} \mathbf{p}_j^{\mathrm{d}} + g_i^{\mathrm{s}} \qquad \forall i \in \mathcal{N} \tag{13b}$$

$$-b_e(\boldsymbol{\theta}_i - \boldsymbol{\theta}_j) - \mathbf{p}_e^{\mathrm{f}} = 0 \qquad \forall e = (i,j) \in \mathcal{E} \tag{13c}$$

$$\underline{\Delta}\theta_e \leq \boldsymbol{\theta}_i - \boldsymbol{\theta}_j \leq \overline{\Delta}\theta_e \qquad \forall e = (i,j) \in \mathcal{E} \tag{13d}$$

$$\boldsymbol{\theta}_{\mathrm{ref}} = 0 \tag{13e}$$

$$\underline{\mathbf{p}_i^{\mathrm{g}}} \leq \mathbf{p}_i^{\mathrm{g}} \leq \overline{\mathbf{p}_i^{\mathrm{g}}} \qquad \forall i \in \mathcal{G} \tag{13f}$$

$$-\overline{S_e} \leq \mathbf{p}_e^{\mathrm{f}} \leq \overline{S_e} \qquad \forall e \in \mathcal{E} \tag{13g}$$

---

The dual DC-OPF problem is stated in Model 5. Dual variables $\lambda^{\mathrm{p}}$ and $\lambda^{\mathrm{pf}}$ are associated to equality constraints (13b) and (13c), respectively. Dual variables $\underline{\mu}^{\theta}, \bar{\mu}^{\theta}$ are associated to lower and upper sides of the voltage angle difference constraint (13d). Finally, dual variables $\underline{\mu}^{\mathrm{pg}}, \bar{\mu}^{\mathrm{pg}}, \underline{\mu}^{\mathrm{pf}}, \bar{\mu}^{\mathrm{pf}}$ are associated to lower and upper bounds on variables $\mathbf{p}^{\mathrm{g}}$ and $\mathbf{p}^{\mathrm{f}}$.

---

**Model 5** Dual of DC-OPF

$$\max_{\lambda, \mu} \quad \sum_{i \in \mathcal{N}} \lambda_i^{\mathrm{p}} \left( g_i^{\mathrm{s}} + \sum_{j \in \mathcal{L}_i} \mathbf{p}_j^{\mathrm{d}} \right) + \sum_{i \in \mathcal{N}} \sum_{j \in \mathcal{G}_i} \left( \underline{\mathbf{p}_j^{\mathrm{g}}} \underline{\mu}_j^{\mathrm{pg}} + \overline{\mathbf{p}}_j^{\mathrm{g}} \bar{\mu}_j^{\mathrm{pg}} \right)$$

$$+ \sum_{e \in \mathcal{E}} \left( \underline{\Delta}\theta_e \underline{\mu}_e^{\theta} + \overline{\Delta}\theta_e \bar{\mu}_e^{\theta} - \bar{s}_e \underline{\mu}_e^{\mathrm{pf}} + \bar{s}_e \bar{\mu}_e^{\mathrm{pf}} \right) \tag{14a}$$

$$\text{s.t.} \quad \lambda_i^{\mathrm{p}} + \underline{\mu}_g^{\mathrm{pg}} + \bar{\mu}_g^{\mathrm{pg}} = c_g \qquad \forall i \in \mathcal{N}, \forall g \in \mathcal{G}_i \tag{14b}$$

$$-\lambda_i^{\mathrm{p}} + \lambda_j^{\mathrm{p}} - \lambda_e^{\mathrm{pf}} + \underline{\mu}_e^{\mathrm{pf}} + \bar{\mu}_e^{\mathrm{pf}} = 0 \qquad \forall e = (i,j) \in \mathcal{E} \tag{14c}$$

$$\sum_{e \in \mathcal{E}_i^+} \left( \underline{\mu}_e^{\theta} - b_e \lambda_e^{\mathrm{pf}} \right) + \sum_{e \in \mathcal{E}_i^-} \left( \bar{\mu}_e^{\theta} + b_e \lambda_e^{\mathrm{pf}} \right) = 0 \qquad \forall i \in \mathcal{N} \tag{14d}$$

$$\underline{\mu}^{\theta}, \underline{\mu}^{\mathrm{pg}}, \underline{\mu}^{\mathrm{pf}} \geq 0 \tag{14e}$$

$$\bar{\mu}^{\theta}, \bar{\mu}^{\mathrm{pg}}, \bar{\mu}^{\mathrm{pf}} \leq 0 \tag{14f}$$

---

PGLearn supports any linear programming solver supported by JuMP. By default, PGLearn solves DC-OPF instances using HiGHS (Huangfu and Hall, 2018).

# B  DATASET FORMAT

This section contains tables describing the format for the case file, the input data files, and the metadata, primal solution, and dual solution files for each of the formulations. Besides the JSON case file, all data is stored in the HDF5 format (The HDF Group, 2024). Each dataset within PGLearn is structured following the diagram below, where `<case>` refers to the snapshot name, `<split>` is "train", "test", or "infeasible", and `<formulation>` is "ACOPF", "DCOPF", or "SOCOPF":

```
<case>
| - case.json
| - <split>
|   | - input.h5
|   | - <formulation>
|   |   | - primal.h5
|   |   | - dual.h5
|   |   | - meta.h5
```

The main data in the `input.h5` files are stored under the `data` key, with metadata (seed numbers and the configuration file used to generate the dataset) stored under the `meta` key. The structure of the input data tables is given in Table 2. The structure of the metadata for each formulation (stored in the `meta.h5` files), is described in Table 3. The reference case data stored in `case.json` is described in Table 7.

Table 2: Input Data Format

| Key | Shape | Meaning |
|---|---|---|
| pd | $(N, |\mathcal{L}|)$ | $\mathbf{p}^d$ – Active power demand |
| qd | $(N, |\mathcal{L}|)$ | $\mathbf{q}^d$ – Reactive power demand |
| branch_status | $(N, |\mathcal{E}|)$ | 0 if branch is disabled, 1 otherwise |
| gen_status | $(N, |\mathcal{G}|)$ | 0 if generator is disabled, 1 otherwise |

Table 3: Metadata Format

| Key | Size | Meaning |
|---|---|---|
| formulation | $(N, 1)$ | Formulation name |
| termination_status | $(N, 1)$ | MOI.TerminationStatusCode |
| primal_status | $(N, 1)$ | MOI.ResultStatusCode for primal solution |
| dual_status | $(N, 1)$ | MOI.ResultStatusCode for dual solution |
| solve_time | $(N, 1)$ | Time spent in solver |
| build_time | $(N, 1)$ | Time spent building JuMP model |
| extract_time | $(N, 1)$ | Time spent extracting solution |
| primal_objective_value | $(N, 1)$ | Primal objective value |
| dual_objective_value | $(N, 1)$ | Dual objective value |
| seed | $(N, 1)$ | MersenneTwister seed |

Table 4: AC-OPF Data Format

| Dual Key | Size | Constraint |
|---|---|---|
| slack_bus | $(N, 1)$ | (7k) |
| kcl_p | $(N, |\mathcal{N}|)$ | (7b) |
| kcl_q | $(N, |\mathcal{N}|)$ | (7c) |
| ohm_pf | $(N, |\mathcal{E}|)$ | (7d) |
| ohm_qf | $(N, |\mathcal{E}|)$ | (7e) |
| ohm_pt | $(N, |\mathcal{E}|)$ | (7f) |
| ohm_qt | $(N, |\mathcal{E}|)$ | (7g) |
| sm_fr | $(N, |\mathcal{E}|)$ | (7h) |
| sm_to | $(N, |\mathcal{E}|)$ | (7i) |
| va_diff | $(N, |\mathcal{E}|)$ | (7j) |
| pg_lb / pg_ub | $(N, |\mathcal{G}|)$ | (7l) |
| qg_lb / qg_ub | $(N, |\mathcal{G}|)$ | (7m) |
| vm_lb / vm_ub | $(N, |\mathcal{N}|)$ | (7n) |
| pf_lb / pf_ub | $(N, |\mathcal{E}|)$ | (7o) |
| qf_lb / qf_ub | $(N, |\mathcal{E}|)$ | (7p) |
| pt_lb / pt_ub | $(N, |\mathcal{E}|)$ | (7q) |
| qt_lb / qt_ub | $(N, |\mathcal{E}|)$ | (7r) |

| Primal Key | Size | Variable |
|---|---|---|
| pg | $(N, |\mathcal{G}|)$ | $\mathbf{p}^{\mathrm{g}}$ |
| qg | $(N, |\mathcal{G}|)$ | $\mathbf{q}^{\mathrm{g}}$ |
| vm | $(N, |\mathcal{N}|)$ | $\mathbf{v}$ |
| va | $(N, |\mathcal{N}|)$ | $\boldsymbol{\theta}$ |
| pf | $(N, |\mathcal{E}|)$ | $\mathbf{p}^{\mathrm{f}}$ |
| qf | $(N, |\mathcal{E}|)$ | $\mathbf{q}^{\mathrm{t}}$ |
| pt | $(N, |\mathcal{E}|)$ | $\mathbf{p}^{\mathrm{t}}$ |
| qt | $(N, |\mathcal{E}|)$ | $\mathbf{q}^{\mathrm{t}}$ |

Table 5: SOC-OPF Data Format

| Dual Key | Size | Constraint |
|---|---|---|
| kcl_p | $(N, |\mathcal{N}|)$ | (11b) |
| kcl_q | $(N, |\mathcal{N}|)$ | (11c) |
| ohm_pf | $(N, |\mathcal{E}|)$ | (11d) |
| ohm_qf | $(N, |\mathcal{E}|)$ | (11e) |
| ohm_pt | $(N, |\mathcal{E}|)$ | (11f) |
| ohm_qt | $(N, |\mathcal{E}|)$ | (11g) |
| sm_fr | $(N, |\mathcal{E}|, 3)$ | (11h) |
| sm_to | $(N, |\mathcal{E}|, 3)$ | (11i) |
| jabr | $(N, |\mathcal{E}|, 4)$ | (11j) |
| va_diff_lb / va_diff_ub | $(N, |\mathcal{E}|)$ | (11k) |
| w_lb / w_ub | $(N, |\mathcal{N}|)$ | (11l) |
| wr_lb / wr_ub | $(N, |\mathcal{E}|)$ | (11s) |
| wi_lb / wi_ub | $(N, |\mathcal{E}|)$ | (11t) |
| pg_lb / pg_ub | $(N, |\mathcal{G}|)$ | (11m) |
| qg_lb / qg_ub | $(N, |\mathcal{G}|)$ | (11n) |
| pf_lb / pf_ub | $(N, |\mathcal{E}|)$ | (11o) |
| qf_lb / qf_ub | $(N, |\mathcal{E}|)$ | (11p) |
| pt_lb / pt_ub | $(N, |\mathcal{E}|)$ | (11q) |
| qt_lb / qt_ub | $(N, |\mathcal{E}|)$ | (11r) |

| Primal Key | Size | Variable |
|---|---|---|
| pg | $(N, |\mathcal{G}|)$ | $\mathbf{p}^{\mathrm{g}}$ |
| qg | $(N, |\mathcal{G}|)$ | $\mathbf{q}^{\mathrm{g}}$ |
| w | $(N, |\mathcal{N}|)$ | $\mathbf{w}$ |
| wr | $(N, |\mathcal{E}|)$ | $\mathbf{w}^{\mathrm{re}}$ |
| wi | $(N, |\mathcal{E}|)$ | $\mathbf{w}^{\mathrm{im}}$ |
| pf | $(N, |\mathcal{E}|)$ | $\mathbf{p}^{\mathrm{f}}$ |
| pt | $(N, |\mathcal{E}|)$ | $\mathbf{p}^{\mathrm{t}}$ |
| qf | $(N, |\mathcal{E}|)$ | $\mathbf{q}^{\mathrm{f}}$ |
| qt | $(N, |\mathcal{E}|)$ | $\mathbf{q}^{\mathrm{t}}$ |

Table 6: DC-OPF Data Format

| Primal Key | Size | Variable |
|---|---|---|
| pg | $(N, |\mathcal{G}|)$ | $\mathbf{p}^{\mathrm{g}}$ |
| va | $(N, |\mathcal{N}|)$ | $\boldsymbol{\theta}$ |
| pf | $(N, |\mathcal{E}|)$ | $\mathbf{p}^{\mathrm{f}}$ |

| Dual Key | Size | Constraint |
|---|---|---|
| slack_bus | $(N, 1)$ | (13e) |
| kcl | $(N, |\mathcal{N}|)$ | (13b) |
| ohm | $(N, |\mathcal{E}|)$ | (13c) |
| va_diff | $(N, |\mathcal{E}|)$ | (13d) |
| pg_lb / pg_ub | $(N, |\mathcal{G}|)$ | (13f) |
| pf_lb / pf_ub | $(N, |\mathcal{E}|)$ | (13g) |

Table 7: Case JSON Format

| Key | Size | Description |
|---|---|---|
| case | – | Snapshot name |
| N | 1 | Number of buses ($\lvert \mathcal{N} \rvert$) |
| E | 1 | Number of edges ($\lvert \mathcal{E} \rvert$) |
| L | 1 | Number of loads ($\lvert \mathcal{L} \rvert$) |
| G | 1 | Number of generators ($\lvert \mathcal{G} \rvert$) |
| ref_bus | 1 | Index of reference/slack bus (1-based) |
| base_mva | 1 | Base MVA to convert from per-unit |
| vnom | $\lvert \mathcal{N} \rvert$ | Nominal voltage |
| pd | $\lvert \mathcal{L} \rvert$ | Reference active power load ($\mathbf{p}^{\mathrm{d}}$) |
| qd | $\lvert \mathcal{L} \rvert$ | Reference reactive power load ($\mathbf{q}^{\mathrm{d}}$) |
| A | $(\lvert \mathcal{E} \rvert, \lvert \mathcal{N} \rvert)$ | Branch incidence matrix in COO format |
| Ag | $(\lvert \mathcal{N} \rvert, \lvert \mathcal{G} \rvert)$ | Generator incidence matrix in COO format |
| bus_arcs_fr | $\lvert \mathcal{N} \rvert$ | Indices of branches leaving each bus ($\mathcal{E}_i$) |
| bus_arcs_to | $\lvert \mathcal{N} \rvert$ | Indices of branches entering each bus ($\mathcal{E}_i^R$) |
| bus_gens | $\lvert \mathcal{N} \rvert$ | Indices of generators at each bus ($\mathcal{G}_i$) |
| bus_loads | $\lvert \mathcal{N} \rvert$ | Indices of loads at each bus ($\mathcal{L}_i$) |
| gs | $\lvert \mathcal{N} \rvert$ | Nodal shunt conductance ($g^{\mathrm{s}}$) |
| bs | $\lvert \mathcal{N} \rvert$ | Nodal shunt susceptance ($b^{\mathrm{s}}$) |
| vmin | $\lvert \mathcal{N} \rvert$ | Voltage magnitude lower bound ($\underline{\mathbf{v}}$) |
| vmax | $\lvert \mathcal{N} \rvert$ | Voltage magnitude upper bound ($\overline{\mathbf{v}}$) |
| dvamin | $\lvert \mathcal{E} \rvert$ | Minimum voltage angle difference ($\underline{\Delta \theta}$) |
| dvamax | $\lvert \mathcal{E} \rvert$ | Maximum voltage angle difference ($\overline{\Delta \theta}$) |
| smax | $\lvert \mathcal{E} \rvert$ | Branch thermal limit ($\overline{S}$) |
| pgmin | $\lvert \mathcal{G} \rvert$ | Minimum active power generation ($\underline{\mathbf{p}^{\mathrm{g}}}$) |
| pgmax | $\lvert \mathcal{G} \rvert$ | Maximum active power generation ($\overline{\mathbf{p}^{\mathrm{g}}}$) |
| qgmin | $\lvert \mathcal{G} \rvert$ | Minimum reactive power generation ($\underline{\mathbf{q}^{\mathrm{g}}}$) |
| qgmax | $\lvert \mathcal{G} \rvert$ | Maximum reactive power generation ($\overline{\mathbf{q}^{\mathrm{g}}}$) |
| c1 | $\lvert \mathcal{G} \rvert$ | Linear cost coefficient |
| gen_bus | $\lvert \mathcal{G} \rvert$ | Bus index of each generator (1-based) |
| load_bus | $\lvert \mathcal{L} \rvert$ | Bus index of each load (1-based) |
| bus_fr | $\lvert \mathcal{E} \rvert$ | From bus index for each branch ($i$) (1-based) |
| bus_to | $\lvert \mathcal{E} \rvert$ | To bus index for each branch ($j$) (1-based) |
| g | $\lvert \mathcal{E} \rvert$ | Branch conductance |
| b | $\lvert \mathcal{E} \rvert$ | Branch susceptance |
| gff | $\lvert \mathcal{E} \rvert$ | From-side branch conductance ($g^{\mathrm{ff}}$) |
| gft | $\lvert \mathcal{E} \rvert$ | From-to branch conductance ($g^{\mathrm{ft}}$) |
| gtf | $\lvert \mathcal{E} \rvert$ | To-from branch conductance ($g^{\mathrm{tf}}$) |
| gtt | $\lvert \mathcal{E} \rvert$ | To-side branch conductance ($g^{\mathrm{tt}}$) |
| bff | $\lvert \mathcal{E} \rvert$ | From-side branch susceptance ($b^{\mathrm{ff}}$) |
| bft | $\lvert \mathcal{E} \rvert$ | From-to branch susceptance ($b^{\mathrm{ft}}$) |
| btf | $\lvert \mathcal{E} \rvert$ | To-from branch susceptance ($b^{\mathrm{tf}}$) |
| btt | $\lvert \mathcal{E} \rvert$ | To-side branch susceptance ($b^{\mathrm{tt}}$) |

## C  DATA LOADING TUTORIAL

PGLearn is compatible with the `datasets` Python library enabling powerful features such as streaming and compatibility with many major ML frameworks. For example, to use the `1354_pegase` dataset, one can run the following code:

```python
from datasets import load_dataset

ds = load_dataset(
    "PGLearn/PGLearn-Medium-1354_pegase",
    split="test",      # train or test
    streaming=True,    # optional; download samples on-demand
    columns=[          # optional; only download some columns
      "input/pd", "ACOPF/primal/pg",
    ]
)

sample = next(iter(ds))
print(len(sample["input/pd"]))         # 673
print(len(sample["ACOPF/primal/pg"]))  # 260

# example torch usage with torch.utils.data.DataLoader
import torch
dl = torch.utils.data.DataLoader(
    ds.with_format("torch"),
    batch_size=8
)
batch = next(iter(dl))
print(batch["ACOPF/primal/pg"].shape)  # torch.Size([8, 260])
```

The HDF5 files can also be downloaded directly from the `script` revision:

```python
from huggingface_hub import snapshot_download

# download the compressed HDF5 files
snapshot_download(
    "PGLearn/PGLearn-Small-14_ieee",
    local_dir="./data",
    repo_type="dataset", revision="script",
    # optional; filter what files to download
    allow_patterns=[
      "*/DCOPF/*", "*input*"
    ],
    ignore_patterns=[
      "*dual.h5.gz", "infeasible/*"
    ],
)

# optional; pre-decompress all files using gzip
from pathlib import Path
import gzip, shutil
for src in Path("./data").rglob("*.h5.gz"):
    dest = src.with_suffix("")
    with gzip.open(src, "rb") as fsrc, open(dest, "wb") as fdest:
        shutil.copyfileobj(fsrc, fdest)
    src.unlink()  # optional; delete the compressed files

# read using h5py
import h5py
h5py.File("./data/train/DCOPF/primal.h5")['pg'].shape  # (756205, 5)
```

