# OpenReview forum: "PGLearn - An Open-Source Learning Toolkit for Optimal Power Flow"
_ICLR.cc/2026/Conference — ICLR 2026 Conference Withdrawn Submission_

### Official Review · Reviewer_McvA · 2025-11-01

**Soundness:** 3
**Presentation:** 3
**Contribution:** 2
**Rating:** 4
**Confidence:** 4

**Summary:**

The paper introduces PGLearn, an open-source suite for machine learning (ML) research on Optimal Power Flow (OPF). It tackles the major bottleneck in this area — the lack of standardized, realistic, and reproducible datasets and evaluation metrics.

**Strengths:**

•	Addresses a key bottleneck in ML for OPF: standardized, reproducible, large-scale data. This dataset will be very useful in ML+OPF researchers.
•	Comprehensive: provides solutions in AC, DC, and SOC formulations, with both primal and dual outputs.
•	Well-designed open-source infrastructure (Julia + PyTorch integration).
•	Realistic data generation that models both correlated and local variations, improving representativeness.

**Weaknesses:**

-	Still synthetic data, but this is understandable given the unavailability of high-fidelity real-world data.
-	Ideally, for a dataset release like this, it would be nice to also accompany it with some ML-OPF benchmarks and report their metrics on the dataset.
-	The intellectual merit is limited. This dataset release does not provide new insights on how ML can be better applied to physical systems, or better NN architectures/training practices for OPF, etc.
- I have doubt on whether this dataset paper is a good fit for ICLR conference. I wonder whether it would be a better fit for conferences with dedicated dataset tracks (e.g. NeurIPS). While ICLR had dataset papers before, those had broad impact across the ML community. The impact of this dataset is limited to the narrow range of researchers in ML+OPF, and the impact for the broader ML community is limited.

**Questions:**

See weakness

---

### Official Review · Reviewer_kPPx · 2025-11-07

**Soundness:** 3
**Presentation:** 3
**Contribution:** 3
**Rating:** 6
**Confidence:** 5

**Summary:**

The reviewer was once the reviewer for the manuscript at ICLR 2025. Since the manuscript does not have significant improvement and modification since then, I will maintain my previous comments and add more questions and rating.

The paper introduces PGLearn, a suite of standardized datasets and evaluation tools designed to advance ML applications in OPF problems. It tries to provide the standard genration of varied data and evaluation metrics by providing realistic datasets that reflect real-world conditions and support multiple OPF formulations.

**Strengths:**

1. The dataset is timely and important given the current active research in ML for OPF
2. The presentation is clear and easy to follow

**Weaknesses:**

1. For my best understanding, the paper mainly focus on the dataset introduction, it does not deliver the technical advances in the search field

2. While the datasets aim to capture real-world conditions, unforeseen variability in actual grids may still limit the applicability of the models trained on these datasets. In addition to the author's previous comments and the testing network size, the variability lies in both the load/demand level but also on the network structure, e.g., the number of nodes, connection, contingency etc.

3. It only focus on OPF problems, which may not be fit for the general audience of the ML community. While I agress the OPF has various formulations and problem structure, its generalization to general OR problem is still limited, e.g., the specific relaxzation, radial topology etc. How could this adapt to or contribute to the existing area is not improved or further discussed in the manuscript.

**Questions:**

1. The authors are suggested to include a general review on the ML for OPF papers, e.g., how ML are generally applied to OPF problems. There have been several literature review papers on it. Including the SOTA model and methods is necessary to improve the quality of the manuscript and including the discussion of how data-driven approach work could help the general audience understand why the dataset and kit is worthwhile.

2. The authors are suggested to state clearly whether the kit contains the function to train/test different ML models? If so, please clarify how to integrate the numerious DNN models and approaches; if not, how should the mentioned Metrics be calculated? Is it still done based on the user side to mannually do so?
Could you provide more details on the "baseline ML models" included in AnonymousRepo2? For example, what specific architectures (e.g., feedforward DNNs, graph neural networks) are supported out-of-the-box, and how does the toolkit handle hyperparameter tuning or integration with popular frameworks like TensorFlow alongside PyTorch?

3. It is also useful to add a set of adversial samples in the dataset for robustness testing under the worst case scenario. The adversial samples here refer to the 'extreme' load inputsinto it, which could lead to solution with high percentage of constraints tightness and lead to the ML model easily makes mistake.

4. Can the suite allow users to self-define the problem and use e.g., API calls, to load the generated dataset? It can be helpful if user have some uncommon reformulated OPF formulations and other general optimization formulation
Regarding custom OPF formulations or even non-OPF optimization problems (e.g., unit commitment or other power system tasks), how does the modular design support extending the dataset generation in AnonymousRepo1? For example, can users define new constraints or objectives via configuration files without altering the core codebase, and is there built-in validation for feasibility?

---

### Official Review · Reviewer_uQ1p · 2025-11-07

**Soundness:** 2
**Presentation:** 2
**Contribution:** 1
**Rating:** 2
**Confidence:** 4

**Summary:**

Following the popularity of ML techniques for solving OPF problems, this paper aims to introduce standardized datasets and evaluation tools for ML and OPF. The main motivation of the paper is addressing data scarcity in ML for OPF research by providing a “data augmentation” strategy given snapshots of electricity networks, such as OPF examples from the PGLib-OPF library. The provided datasets also support AC, DC, and second-order cone formulations of OPF, and include time-series data.

**Strengths:**

Like most prior work in the field, PGLearn uses a sampling scheme to convert static snapshots into datasets of OPF instances. For example, as shown in Algorithm 1, one knows exactly the probability distribution generating the datasets, and can therefore recreate the same dataset. This sets the issue apart from the classical ML reproducibility problem, where we do not know the true data distribution generating image, text, or other data. Therefore, I do not think the authors overcome a very high barrier for democratizing and accelerating research and innovation in machine learning applications for optimal power flow problems, as they claim.

However, solving the second-order conic relaxations of large datasets in Table 1 for many OPF instances could potentially take a lot of time. In that sense, it could be useful that the library provides these solutions.

**Weaknesses:**

This paper should be rejected because it does not go beyond simply releasing data. ICLR is not a “dataset-only” venue; papers must show scientific contributions, insights, or methodological novelty. Also, the paper could use some polishing.

- The provided sampling methods either have marginal effect on capturing network-level load variability (Demand Sampling) or are repetitions of other works (Status and Time-Series Sampling). Furthermore, it is not convincing that their datasets actually provide this variability. The main motivation of the paper is the lack of (open) datasets capturing variation of load profiles in real-life operating conditions. The authors address this problem by, as they write, “explicitly capturing both global and local variability in the data generation,” as opposed to other works simply adding random perturbations to each bus demand. However, their Demand Sampling method simply multiplies perturbed active/reactive demand at each bus by the same constant $b$ sampled from a uniform distribution. While this contribution is quite marginal, it also reduces the approximate correlation between bus demands to a linear relationship, which is not realistic for power systems. This load profile would only be realistic if the support of  $b$  is around 1.

**Questions:**

- Line 307 mentions N-1 contingencies. Are you solving any security-constrained OPF problems? If not, what is the purpose of this part? Is it only for creating slightly different topologies?
- The abstract suggests that the paper “for the first time, includes time-series data for several large-scale systems.” But it then states, “Motivated by this mismatch, the Texas7k and Midwest24k cases include one year of synthetic time-series data at an hourly granularity.” This makes it seem like the paper just interpolates these data. Could you clearly state the contribution here?
- Do the generated OPF datasets include any variations of generator cost curves, or upper/lower bounds on branch flows, generation, or bus voltage magnitudes?

---

### Note · Authors · 2025-11-24

I have read and agree with the venue's withdrawal policy on behalf of myself and my co-authors.